# Effects of Infrared-Assisted Refractance Window™ Drying on the Drying Kinetics, Microstructure, and Color of *Physalis* Fruit Purée

**DOI:** 10.3390/foods9030343

**Published:** 2020-03-16

**Authors:** Luis Puente-Díaz, Oliver Spolmann, Diego Nocetti, Liliana Zura-Bravo, Roberto Lemus-Mondaca

**Affiliations:** 1Departamento de Ciencia de los Alimentos y Tecnología Química, Facultad de Ciencias Químicas y Farmacéuticas, Universidad de Chile, Santos Dumont 964, Independencia, Santiago 1058, Chile; oliverspolmann@ug.uchile.cl (O.S.); lzura@userena.cl (L.Z.-B.); 2Departamento de Tecnología Médica, Facultad de Ciencias de la Salud, Universidad de Tarapacá, Arica 1000665, Chile; dnocetti@uta.cl

**Keywords:** *Physalis*, refractance window, infrared, diffusivity, modelling, microstructure, color

## Abstract

The objective of this work was to study the influence of the drying temperature, infrared (IR) radiation assistance, and the Mylar™ film thickness during *Physalis* fruit purée drying by the Refractance Window™ (RW™) method. For this, a RW™ dryer layout with a regulated bath at working temperatures of 60, 75, and 90 °C, Mylar™ thicknesses of 0.19, 0.25, 0.30 mm and IR radiation of 250 W for assisting RW™ drying process was used. Experimental curves data were expressed in moisture ratio (MR) in order to obtain moisture effective diffusivities (non-assisted RW™: *D_eff_* = 2.7–10.1 × 10^−10^ m^2^/s and IR-assisted RW™: *D_eff_* = 4.2–13.4 × 10^−10^ m^2^/s) and further drying curves modeling (Page, Henderson–Pabis, Modified Henderson–Pabis, Two-Term, and Midilli–Kucuk models). The Midilli–Kucuk model obtained the best-fit quality on experimental curves regarding statistical tests applied (Coefficient of Determination (*R*^2^), *Chi-Square (χ*^2^) and Root Mean Square Error (*RMSE*). Microscopical observations were carried out to study the RW™ drying conditions effect on microstructural changes of *Physalis* fruit purée. The main findings of this work indicated that the use of IR-assisted RW™ drying effectively accelerates the drying process, which achieved a decrease drying time around 60%. Thus, this combined RW™ process is strongly influenced by the working temperature and IR-power applied, and slightly by Mylar™ thickness.

## 1. Introduction

*Physalis* (*Physalis peruviana* L.), goldenberry, or uchuva is a berry that has a high content of vitamin A, B, and C, β-carotene, phosphorus, iron, and bioactive compounds. It also has antioxidant activity, anti-hepatotoxic activity, anti-proliferative effects in hepatoma cells, anti-inflammatory and antidiabetic activity, so that their consumption provides health benefits and reduces the risk of acquiring certain diseases, such as cancer, malaria, asthma, hepatitis, dermatitis, and rheumatism, for which *Physalis* fruit, it is considered as a promising functional food [1].

On the other hand, dehydration by the Refractance Window™ (RW™) drying method is a drying technology considered as emerging for the drying of food material that is used to evaporate water from liquid or semi-liquid foods [2]. This drying technique is an indirect contact technique, in which food material is heated by both conduction and radiation as heat transfer mechanisms, where the thermal energy coming from hot water is transferred through an infrared transparent polyethylene terephthalate film, known commercially as Mylar™, which is placed between the food material and the water. This plastic film having the property to be able to transfer heat by conduction and by radiation, allows the food material to dry at moderate temperatures achieving usually shorter drying times. The RW™ method is attractive to the heat-sensitive food industry due to the high quality of the products obtained, and that the equipment is relatively cheap (one-third compared to a freeze-dryer). It also has the advantage that the operational cost of a dryer by the RW™ method is approximately half that of a freeze-dryer [3].

Traditional drying technologies, such as hot-air convective drying, vacuum drying, microwave drying, freeze-drying, among others, aim to ensure that the food quality and stability are preserved during the storage period, being, at the same time, as efficient as possible. However, so as to fulfill efficiently the drying process, shortening operation times and enhancing product quality, as well as using combined drying technologies in which different sources of energy are involved, are becoming more and more current. In this sense, there is the possibility of assisting the different drying processes with several forms of energy that accelerate transfer of matter from the diverse feed sources. Infrared (IR) radiation assistance could be seen as a convenient and alternative method to combine with RW^TM^ drying process of high quality fruit pulps or purées. Hence, when food materials are subjected to an electromagnetic field, the wave penetrates directly into the material, resulting in fast volumetric heating. These electromagnetic waves in IR radiation are displayed under three categories ranges, 0.78–1.4 μm for the near-infrared (NIR), 1.4–3.0 μm for the middle-infrared (MIR), and 3.0–1000 μm for the far-infrared (FIR). Quick energy absorption by water molecules causes rapid evaporation of water, creating an outward flux of rapidly escaping water vapor. In this way, it can reduced drying times, minimize quality losses, and achieve uniform heating, among other advantages [4].

Among the most important quality properties of dried food, color and texture are widely noteworthy [5]. The food surface color is the first quality parameter evaluated by consumers, and is critical in product acceptance, even before it is tasted. The CIELab system has been widely used to describe color changes during food processing, where this system has been related to the types and quantities of some components present in those products, e.g., carotenoid content. Whatever the drying processes, the dried food material requires microscopical analysis to address structural changes produced at the cellular level. These changes are related to the loss of water from the inner parts towards the surface and the surrounding air, possibly causing stiffness, spoilage, and disruption of the cell walls, or even a collapse of the cell tissues. The structural properties of the product depend on the type of drying technique, operative variables, and food microstructure formed during the process [6].

In order to successfully transfer knowledge acquired experimentally from studies on food dehydration into industrial applications, drying kinetics modeling is necessary. Besides, a mathematical model is an important tool used to optimize the management of operating parameters and to simulate the drying processes behavior. These simple models, also known as thin-layer models, allow prediction of mass transfer during dehydration and are applied to simulate drying curves under similar conditions [7]. 

In the light of the above considerations, as RW™ drying method is considered an emerging technique based on a thin-layer configuration of the food materials, along with considering that the use of infrared radiation to accelerate/assist drying processes, has proved to be effective and easy to implement. The main advantage of combined drying technologies is being able to dehydrate food materials using high temperatures and short drying time, which allows attaining high-quality retention and/or incorporate a high added-value to food products [3,5]. Therefore, the main aim of this work was to study the influence of RW™ drying method on drying behavior, microstructural aspects, and color changes of *Physalis* fruit purée under different drying temperatures, Mylar™ film thickness, and with/without the assistance of IR radiation.

## 2. Materials and Methods

### 2.1. Raw Material

*Physalis* fruits were purchased from local producer Hortifrut Ltd., Santiago, Chile. The samples were selected to provide a homogenous group, based on the date of harvest, color, size, and freshness, according to visual analysis. This visual inspection consisted of discard fruits with mechanical damage, cuts, perforations, and impacts. Then, the fruits were washed in tap water to eliminate any surface contamination of foreign matter, and refrigerated immediately (5.0 ± 0.2 °C). Before drying experiments, the *Physalis* fruit purée was prepared using a food blender (Minipimer MR-4000) in order to obtain a homogeneous sample containing pulp, seeds, and peel. Afterwards, the *Physalis* purée samples (200 ± 2 g, approximately) were packaged in low density polyethylene bag (10 × 15 cm), vacuum heat-sealed, and kept under freezing conditions (−18.0 ± 0.2 °C), until drying process. The samples frozen were thawed at 5.0 ± 0.2 °C. The moisture content was determined by AOAC (Association of Official Analytical Chemists) method no. 934.06 (AOAC, 1990), employing a vacuum oven (Mlab Scientific MVO-024, ThermoFisher Scientific, Waltham, MA, USA) at 70 °C for 72 h, and an analytical balance accurate to 0.0001 g (Jex120, CHYO, Tokyo, Japan). Moreover, the total soluble sugar content was calculated with an Abbe refractometer (ATAGO, 1-T, Japan) and the water activity was determined at 25 °C by a Novasina water activity instrument (model TH-500, Pfaffikon, Switzerland). 

### 2.2. RW^TM^ Drying Experiments

A laboratory-scale RW™ drying system was built using a 30 L thermo-regulated bath using temperatures of 60, 75, and 90 °C and a set of Mylar™ film (polyester type D, DuPont, Wilmington, DC, USA) with different thicknesses of 0.19, 0.25, and 0.30 mm. Three containers were built from Mylar™ film; these have an effective transfer area of 140 cm^2^ where Physalis purée was placed and then spread over the Mylar™ film aided by a spatula blade with a 3 mm gap. A water level control device was used to keep a permanent contact between the water and the Mylar™ film. The assistance with IR radiation was obtained by using an infrared lamp (General Electric Lighting, GE Company, Cleveland, OH, USA) of 250 W (wavelength: 700–1200 nm, 120 Volts, and 60 Hertz), which was placed in a way that ensured covering the greatest surface portion of spread *Physalis* purée over the Mylar™ film (various previous assays enabled an adequate IR lamp placement), as can be seen in Figure 1. Both non-assisted RW™ and IR-assisted RW™ drying experiments, the weight changes on *Physalis* purée were measured using a digital scale (Radwag AS 220-R2, Torunska, Poland) until reaching an approximate moisture of 10% (w.b.) during each drying experiment under study. The dried samples were stored in polyethylene bags at (5.0 ± 0.2 °C), vacuum heat-sealed, protected from sunlight, until further microstructure and color analysis. All drying experiments were performed in triplicate and the results were expressed as the mean ± standard deviation.

### 2.3. Moisture Effective Diffusivity and Drying Modeling 

This value resulted from predicting *Physalis* purée drying behavior that takes place during the falling rate period, during which water is transported to the surface material by diffusion transport phenomena being necessary the use of Fick’s second diffusion law (Equation (1)). Thus, moisture ratio (*MR*) is the dependent variable, which relates the gradient of the sample moisture content in real-time to both initial and equilibrium moisture content. Experimental data were expressed in terms of the *MR* and were calculated according to Equation (2). Then, the integrated equation of Fick’s second law was also used for long time periods and thin-layer in one dimension, representing the first term in the development of the series, from which the diffusional coefficient is obtained for each temperature. The moisture effective diffusivity (*D_eff_*) was calculated according to Equation (3).
(1)$$\frac{\partial MR}{\partial t}=D_{eff}\;\frac{\partial^{2}MR}{\partial x^{2}}$$
(2)$$MR=\frac{X_{t}-X_{e}}{X_{0}-X_{e}}$$
(3)$$MR=\frac{8}{\pi^{2}}exp\;\left(-\frac{\pi^{2}\;D_{eff}\;t}{4L^{2}}\right)$$
where *MR* is moisture ratio (dimensionless); *D_eff_* is the moisture effective diffusivity (m^2^/s); *t* is drying time (s); *x* the spatial dimension (m); *X_t_* is the moisture content (g water/g dry matter, d.m.) at any time; *X_o_* is initial moisture content (g water/g d.m.); *X_e_* is the equilibrium moisture content (g water/g d.m.), and *L* is the half-thickness (m) of the *Physalis* purée thin-layer on the Mylar™ film. 

Diverse mathematical models have been proposed to predict the drying curves behavior of food and agricultural products. The most common drying models are listed in Table 1.

### 2.4. Activation Energy and Arrhenius Factor

The activation energy was obtained from the Arrhenius equation, where the temperature dependence of *D_eff_* was calculated by the Arrhenius equation (Equation (4)) described by [8]:(4)$$D_{eff}=D_{o}\exp\left(-\frac{E_{a}}{RT}\right)$$
where *D_o_* is a pre-exponential factor for the Arrhenius equation (m^2^/s), *E_a_* is activation energy (J/mol), *R* is gas universal constant (8.3145 J/mol K) and *T* is drying temperature (K).

### 2.5. Electronic Microscopy

To perform the scanning electron microscopy (SEM), *Physalis* purée samples, from the drying experiments, were fixed with 2.5% glutaraldehyde in 0.1 M sodium cacodylate buffer for 2 h, after this period they were washed 3 times × 5 min in double distilled water. Subsequently, the samples were (1) washed with 0.1 M × 5 min sodium cacodylate buffer, (2) dehydrated in ascending alcohols 50, 70, 95, 100 °I and 100 °II for 5 min each, and (3) dried until the critical point, by placing the samples in the Autosamdri-815 Critical Point Dryer for 30 min. This process allowed to remove all water molecules through CO_2_ blast and ensured the maximum structural preservation for the samples. Finally, the samples were mounted on Aluminum holders and metalized with gold in denton vacuum desk metallizer. Microphotographs were obtained using a scanning electron microscope (Jeol brand JSM IT300LV) operating at 15.0 kV.

### 2.6. Color Measurements

For the color analysis, a 3nh colorimeter (model NR110, Shenzhen Sanenshi Technology Co., Ltd., Shenzhen, China) was used. Color changes were expressed in CIELab system coordinates, *L** (whiteness/darkness), *a** (redness/greenness), and *b** (yellowness/blueness), standard illuminant D_65_, and observer 10°. The above parameters were measured for fresh and dried samples [9]. Then, the colorimeter yielded values for each spot, which were converted to total color difference value (*∆E*) using Equation (5), where *L_o_*, *a_o_*, and *b_o_* are the control values for fresh *Physalis* purée. Ten replicate measurements were performed and results were expressed as the mean ± standard deviation.
(5)$$\Delta E=\sqrt{{\left(L^{*}-L_{o}\right)}^{2}+{\left(a^{*}-a_{o}\right)}^{2}+{\left(b^{*}-b_{o}\right)}^{2}}$$

### 2.7. Statistical Analysis

An ANOVA analysis was performed for diffusivities and color parameters values, obtained by StatGraphics Plus v.5.1 (Statistical Graphics Corp., Herndon, VA, USA), including a MRT (multiple range test) to recognize significant differences between Mylar™ film thicknesses, applied temperatures and difference in RW^TM^ methods (with or without IR radiation). As to mathematical models of drying to be evaluated, a regression analysis was carried out. The determination coefficient (*R*^2^; Equation (6)) was the first criterion to select the best model to describe the drying kinetics behavior. Moreover, Chi-square (*χ*^2^; Equation (7)) and Root Mean Square Error (RMSE; Equation (8)) were calculated to evaluate the fit quality of the models to experimental data. Where, the *R*^2^ > 0.85, *χ*^2^ << 0.1 and *RMSE* << 0.1, were selected as optimal criteria in order to evaluate the fit quality of the proposed models.
(6)$$R^{2}=1-\frac{\sum_{1}^{N}{\left(MR_{exp,i}-MR_{pre,i}\right)}^{2}}{\sum_{1}^{N}{\left({\bar{MR}}_{exp,i}-MR_{pre,i}\right)}^{2}}$$
(7)$$\chi^{2}=\frac{\sum_{i=1}^{N}{\left(MR_{exp,i}-MR_{pre,i}\right)}^{2}}{N-z}$$
(8)$$RSME=\sqrt{\frac{\sum_{i=1}^{N}{\left(MR_{pre,i}-MR_{exp,i}\right)}^{2}}{N}}$$
where: *MR_exp,i_* is the experimental moisture content; *MR_pre,i_* is calculated moisture content; *i* is the number of terms; *z* is a constant number and *N* is the number of data.

## 3. Results

### 3.1. Drying Experiments

The initial moisture content, sugar content and water activity of *Physalis* purée were of 4.155 ± 0.012 g water/g d.m, 15.48 ± 1.25 °Brix, 0.975 ± 0.001, respectively. Regarding drying kinetics, all the curves presented an exponential decay behavior and the influence of drying temperature is again remarkable (Figure 2). The same tendency has been presented in other reports [10,11,12,13], where a decrease in process time and an increase in drying rate occurred with an increase in the drying temperature. While the drying times to reach equilibrium moisture content around 10% varied from 75 to 180 min. From the experimental drying curves of the non-assisted RW™ method, the shorter drying times were obtained at a temperature of 90 °C and a Mylar™ thickness of 0.19 mm and the longer times using a Mylar™ thickness of 0.29 mm and a drying temperature of 60 °C. Whereas when IR-assisted RW™ was used, the drying curves were more pronounced compared with just non-assisted RW™; a similar behavior was also observed in infrared-assisted convective drying of murta berry [7]. 

Finally, the difference between drying temperatures and applied methods shows a significant decrease in drying time, with a difference of up to 100 min, that is the case of non-assisted RW™: 160 min at 60 °C and IR-assisted RW™: 60 min at 90 °C, both using Mylar thickness of 0.19 mm. Therefore, the temperature difference caused a decreasing drying time of around 20% to 30% in non-assisted RW™ and IR-assisted RW™, while the addition of the IR radiation led to a decreased RW™ drying process time of up to 62%, approximately. Against that background, Vega-Gálvez et al. [14] presented drying time between 300 and 800 min for air-convective drying of *Physalis* fruit in the 60 to 90 °C temperature range. Furthermore, Cabrera Ordoñez et al. [15] evaluated the influence of two temperatures (50 and 60 °C) and two air velocity (2.0 and 3.0 m^2^/s) on *Physalis* drying, where drying times between 17 and 44 h were reached. Junqueira et al. [16] studied the effect of diverse pretreatments and convective drying on kinetics and quality parameters of *Physalis* fruit. The authors showed the drying times from 450 to 650 min, for each of the experiments.

RW™ drying has shown to be a promising drying technology, mainly in time reduction as compared with conventional dehydration methods. Furthermore, when using both combined drying technologies, IR and RW™, for industrial-scale dehydration of fruit and vegetable purée and pulp, as well as other food material, these could be considered as a synergy emerging technology, which allows reducing the drying time and thereby energy consumption.

### 3.2. Moisture Effective Diffusivities

Effective diffusivities values were in the range of 2.74 to 13.40 × 10^−10^ m^2^/s, in Table 2 it can be observed that as the process temperature raises, there is a consequent increase in the effective diffusivity, and this behavior and order of magnitude has been reported for other RW™ drying methods, such as reported [10] for mango flakes (*D_eff_*: 5.43–9.33 × 10^−10^ m^2^/s); [11] apple slices (*D_eff_*: 2.5–7.14 × 10^−10^ m^2^/s); [12] apple slices (*D_eff_*: 1.25–14.3 × 10^−10^ m^2^/s), salmon tissue (*D_eff_*: 1.24–3.09 × 10^−10^ m^2^/s) and beef slices (*D_eff_*: 1.25–3.86 × 10^−10^ m^2^/s); [17] mango pulp (*D_eff_*: 7.0–11.0 × 10^−9^ m^2^/s); [18] yam slices (*D_eff_*: 1.94–2.54 × 10^−9^ m^2^/s); [19] papaya purée (*D_eff_*: 4.25–7.68 × 10^−10^ m^2^/s), and [20] cucumber fruit slices (*D_eff_*: 1.94–2.54 × 10^−9^ m^2^/s). It needs to be mentioned that *D_eff_* values of previous researches varied because of different drying temperature, sample moisture content, application of pre-treatments and Mylar™ thickness. *D_eff_* values studied for similar conditions of temperature and thickness (with difference in infrared assistance) indicate significant differences due to the inclusion of the IR radiation that increases the moisture diffusivity values in all the compared experiments, which implies that the addition of the IR radiation directly and significantly influences the migratory capacity of the water that is extracted in the RW™ drying process.

In both treatments (non-assisted RW™ and IR-assisted RW™), the effective diffusivity values varies significantly with the used temperature, for it, an ANOVA is necessary to realize. Thus, the ANOVA performed on *D_eff_* values showed a significant influence of drying temperature on this physical parameter (*p*-value < 0.05). Moreover, a MRT to determine the significant means among the *D_eff_* was delivered, showing three homogeneous groups for each Mylar™ thickness used. This temperature-dependence effect has been described using Arrhenius-type relationship to get the activation energy (*Ea*) from the effective diffusivity data. Then, the same ANOVA carried out on *D_eff_* values presented a significant influence of Mylar™ film thickness on the diffusivity values, keeping constant the RW™ method (*p*-value < 0.05). MRT analysis didn’t show a greater difference among the *Deff* values vs Mylar™ thickness, showing two homogeneous groups for each drying temperature used (non-assisted RW™ drying → 60 °C: 0.19 mm and 0.25–0.29 mm/75 °C: 0.19–0.25 mm and 0.29 mm/90 °C: 0.19–0.25 mm and 0.19–0.29 mm; IR-assisted RW™ drying → 60 °C: 0.19–0.25 mm and 0.29 mm/75 °C: 0.19 mm and 0.25–0.29 mm/90 °C: 0.19–0.25 mm and 0.29 mm). This showed a clear effect of drying temperature but only a weak effect of Mylar™ film thickness. In this way, it would be possible to select only one appropriate Mylar™ film thickness with respect to the kind of dried food to be developed. Furthermore, this study confirmed again that temperature is one of the main drying operational variables for any drying method, along with IR-assistance also affecting significantly the drying process. Thus, information obtained also can be important for the next hybrid dryer performance taking into account food quality as well as economic costs [12].

Furthermore, a third ANOVA assessment was established, in this case, the *D_eff_* values were evaluated considering the same Mylar™ thickness but different RW™ drying method. From this analysis obtained, it was demonstrated a significant influence of assistance of the infrared system on the increased *D_eff_* (*p*-value < 0.05), in all cases evaluated (x,y). This showed a clear influence of IR radiation as an assistance system for any drying method. Likewise, diverse studies have reported that moisture diffusivities under infrared-assisted drying operations were higher compared to those of non-assisted drying process [21,22,23]. This could be because, during IR-assisted RW™ heating of the moist *Physalis* purée, IR radiation impinged on the exposed spread purée and penetrated it. This, radiation energy is readily converted into heat and the purée exposed to IR radiation is intensely heated, reaching a temperature gradient almost to zero within a short period [21].

### 3.3. Kinetic Parameters

The kinetic parameters of empirical models can be observed in both RW™ method without IR and IR-assisted RW™ drying method in Table 3 and Table 4, respectively. The parameter *k*, *k_o_*, and *k*_1_ values increase as drying temperature increases, this behavior is observed for most of the models and for each RW™ drying process. In addition, it can be seen as the inclusion of the IR radiation on RW^TM^ drying method caused that the parameters *k*, *k_o_*, and *k*_1_ were also directly dependent on the heat source (IR radiation), which indicates that the linear increase of parameters *k* would be related to the physical parameter, *D_eff_* [24]. The other empirical parameters (*n*, *a–c*) did not present significant variations despite increasing the temperature, this could occur due to the food matrix structure that is dried. Furthermore, some authors have found the n parameter not to vary with the temperature, as well as showed an air velocity linear dependence of this parameter *n* [25,26]. The authors [27] proposed that particularly for parameter *n*, this empirical parameter could be affected in the case of drying fruit slices with or without skin, increasing accordingly the shell thickness. Since this study presented a fruit purée drying by RW™ method, these empirical parameters would not be affected by the presence of the shell, despite the temperature being increased, as well as IR radiation included. On this basis, the Mylar™ film thickness could have been an RW™ drying process operational parameter, which could have affected to empirical parameters, but this behavior did not happen.

Statistical evaluation for each mathematical model showed a good fit because low 0.002 ≤ *RMSE* ≤ 0.116 and 0.001 ≤ *χ*^2^ ≤ 0.024 values were obtained. Furthermore, taking account the criterion of *R*^2^ ≥ 0.85, most of the models displayed a good fit quality to all experimental data. However, regarding the statistical values obtained, Midilli–Kucuk (0.005 ≤ *RMSE* ≤ 0.012; 0.001 ≤ *χ*^2^ ≤ 0.004 and 0.997 ≤ *R*^2^ ≤ 0.999) and Page (0.002 ≤ *RMSE* ≤ 0.021; 0.001 ≤ *χ*^2^ ≤ 0.003 and 0.991 ≤ *R*^2^ ≤ 0.999) models showed the best-fit quality on experimental data, for non-assisted RW™ and Infrared-assisted RW™ drying, respectively. Some researchers have attained good outcome when applying both Midilli–Kucuk and Page models on the drying kinetics of different food subjected to different hybrid drying method [9,28,29] as well as the RW™ drying method [12,13]. Always bearing in mind the use of the experimental data or any of the developed models to calculate drying time to reach equilibrium moisture content gives results that differ slightly in magnitude.

### 3.4. Activation Energy

The activation energy values calculated based on *D_eff_* values for the temperature range 60 to 95 °C, taking account both RW™ method and Mylar™ thickness as independent-factors. The *E_a_* values for non-assisted RW^TM^ drying methods were of 41.3, 33.4, and 31.1 kJ/mol for each Mylar™ thickness of 0.19, 0.25, and 0.29 mm, respectively. Whereas *E_a_* results for IR-assisted RW™ drying technique were of 34.5, 33.0, and 26.6 kJ/mol for each Mylar^TM^ thickness of 0.19, 0.25, and 0.29 mm, respectively. From comparing the *Ea* results obtained by different RW™ methods for the same Mylar thickness, the non-assisted RW™ drying method obtained *Ea* values slightly higher than IR-assisted RW™. This may be in part because *Ea* is the energy barrier to activate water diffusion; hence, the IR-assisted RW™ would need less energy to start the moisture diffusion transport. The results obtained are similar to those obtained in other report [11] on the air drying of *Physalis* fruit obtaining 38.78 kJ/mol, other results obtained are similar, 37.27 kJ/mol for figs [27]; but significantly less than those obtained [30] for the Berberis between 110.84–130.61 kJ/mol.

### 3.5. Microstructure Analysis

It’s worth recalling that fresh *Physalis* purée is conformed of flesh, peel, and seeds. Hence the microstructural observations were accomplished on these different sections of fresh *Physalis* purée. The microstructure analysis consists of a complex network made of peel and seeds together with the flesh from the fruit, holding water about them. 

Figure 3A shows the microphotography of an integral peel piece from fresh *Physalis* purée. This microphotography enables exhibiting a skin piece very properly for observation. From the microphotography, a kind of bumps, called “emergencies” by the author [31] is shown, which has been described extensively by [32]. These emergencies behave as a secretory structure and lend the fruit a kind of semi-rough texture, in addition, these are formed by cells of the epidermis and hypodermis. Concerning Figure 3B, this shows the seed pieces with a subtle differentiation between the test (outer layer of the seed) and the parenchymal tissue (which functions as a support in the cell network) described by [31]. Beyond peel pieces “emergencies” size, differences in the homogeneity were observed, where seed microstructure displayed a more homogeneous microstructure. 

From Figure 4, it is possible to observe the effect of both non-assisted RW™ and IR-assisted RW™ drying processes on the *Physalis* purée samples. Figure 4A shows that the cellular network belonging to the peel was affected by the extraction of free water during the drying process, the same situation occurs in Figure 4B, increasing the trend and observing a rough surface. As to Figure 4C, it was not possible to clearly observe the surface described above. It was just possible to recognize peel tissue due to the “emergencies” that were described above, where the cellular damage obtained could have been caused by the application of high drying temperature, in this case at 90 °C (both RW™ drying methods); hence, obtaining a more aggressive evaporation of the water available at the cellular level, breaking the walls of the cells present in the flesh and peel. This fact may also be related to different local chemical composition (e.g., soluble solids, pH, ionic strength, specific cations) that can influence the dried *Physalis* purée deformation and structure.

Figure 4D–F shows dried *Physalis* purée peel by IR-assisted RW™ drying method, where these drying experiments maintained the tendency to compare it with the fresh *Physalis* purée peel (Figure 3A) obtaining a rough surface, with an evident loss of water from the cells. This could be present with a more marked tendency than in those obtained in Figure 4A–C. In the same way this can also be observed when comparing the same drying temperatures with and without the use of infrared energy. Figure 4A shows a dried peel with a smoother surface than Figure 4D, indicating in both situations described above the incidence of infrared assisted RW™ drying. Likewise, the tendency towards rough surfaces is maintained in Figure 4E, also observing the emergencies described above. Finally, in Figure 4F, as in Figure 4C, the cellular damage is higher than that observed in Figure 4C, this may occur because infrared assistance, including the drying process with a higher temperature, raises the internal temperature of the extracted water more aggressively, this being an explosive process, damaging the observed tissues.

Figure 5 presents the observations performed on dried *Physalis* purée seeds, where it is possible to observe as the seed structure changes slightly as the drying temperature, which can be seen in both RW™ drying methods. This change was expressed in its smooth surface towards a rougher one, obtaining a better differentiation of the parenchymal tissue, it is also possible to observe that the IR-assisted RW™ drying caused an increase in the water extraction. In addition, the quality of the end products by RW™ is higher as compared with conventional drying methods [12].

### 3.6. Color Parameters and ∆E

The colorimetric values showed that all *Physalis* purée samples lost *L** coordinate, got closer to the green color (*a** coordinate) and also to the blue color, losing yellow tones (*b** coordinate), regarding fresh *Physalis* samples. Colorimetric coordinates of fresh *Physalis* purée samples were 83, 99, 34, 62, and 72, 4 to *L**, *a** and *b**, respectively. 

The *L** coordinate values (Table 5) decreased by both RW™ drying methods, this is frequent in most drying methods, however, particularly in RW™ drying, there are some similar results obtained in strawberries [33] and also in mango flakes [34]. There are some works presenting that the higher the degree of browning, the lower the sample *L** coordinate value. These chemical reactions are a consequence of reducing sugars and amino acids in the food being dried, where particularly in *Physalis* fruit that has different amino acids (leucine, lysine, and isoleucine) and sugars (sucrose, glucose, and fructose). Thereby, dried *Physalis* purée would become darker, probably because of an extensive Maillard reaction, as in this particular case of IR-assisted RW™ treated *Physalis* purées.

Regarding *a** coordinate (Table 6), diverse differences have been seen according to the food to be dried. It was found that asparagus mash drying kept the green color better than any other type of drying [35]. On the other hand, a significant decrease of the *a** coordinate has been reported in work on the Refractance Window™ drying of strawberries [33]. Finally, the decrease in *b** coordinate (Table 7) could be an indication of the loss of carotenoids present in dried *Physalis* purée, unlike the results that have been reported in carrot flakes drying [36], which have improved the retention of these compounds. Furthermore, the effect of decreasing water activity in amorphous dehydrated systems can be decisive on the non-enzymatic Maillard browning reaction rate and formation of brown pigments [13]. 

ANOVA analysis showed that drying temperature (a, b, c) had a significant influence on color indices of *L** and *b** value (*p*-value < 0.05). Moreover, although slice Mylar™ thickness (A, B, C) influenced *L** and *b** values substantially, it failed to affect *a** values considerably. Ochoa-Martinez [7] demonstrated that the film thickness had significant effects on *L**, *a**, and *b** coordinates for dried mango slices by Refractance Window™ drying. Moreover, a new ANOVA analysis found that RW™ drying method had a significant effect on *L**, *a**, and *b** value (*p*-value < 0.05) but only at 90° (x, y, z).

As to the total color difference or *ΔE* as shown in Figure 6, for both RW™ drying conditions, highly significant differences were observed in dried *Physalis* purée (between 50 ≤ *ΔE* ≤ 75). An increase in the drying temperature when processing without infrared resulted in an increase in Δ*E*. In general, the changes of *ΔE* are mainly due to changes in the chromatic parameters *a** and *b**, where Abonyi et al. [33] found in strawberries and carrots and Nindo et al. [35] presented in asparagus with *ΔE* values between 20 and 25. Fisher et al. [37] and Busso et al. [38] also found similar color changes after the thermal treatment of pomegranate juice and blackcurrant, maqui berry, and blueberry pulps, where they ascribed it to anthocyanin losses.

Thus, color changes caused by both RW™ drying process could have occurred, not only by the non-enzymatic browning reaction but also by the destruction of the pigment present in the *Physalis* purée, such as β-carotene. Nevertheless, the addition of the IR radiation could suppose a very invasive thermal shock for the *Physalis* purée drying, obtaining differences of higher color than with respect to the non-assisted RW™ drying process (e.g., IR-assisted RW™ drying at 90 °C: 65 ≤ *ΔE* ≤ 75). That is why, since food surface color is one of the quality parameters most evaluated by consumers, and is critical in the acceptance of the product, new strategies should also be considered for developing dried fruit purée, such as pre-treatments with additives that could better protect color compounds. 

Then, an ANOVA assessment was established, where the *ΔE* values were evaluated considering the diverse factors under study, drying temperature, Mylar™ thickness, and RW™ drying method. This ANOVA analysis found a significant influence of drying temperature as well as RW™ drying method on the *ΔE* values (*p*-value < 0.05), in all cases evaluated. Whereas, Mylar™ thickness didn’t show a clear influence on the *ΔE* values (*p*-value > 0.05), in the same experiments evaluated.

## 4. Conclusions

RW™ drying assisted by infrared heating (IR-assisted RW™) is a method that allows moisture reduction of *Physalis* purée samples in relatively shorter time than non-assisted RW™ and other traditional drying methods. The drying kinetics behavior was strongly influenced by the drying temperature and the IR-assistance, whilst slightly by Mylar™ thickness. Both the *D_eff_* (non-assisted RW™: 2.7 × 10^−10^ ≤ *D_eff_* ≤ 10.1 × 10^−10^ m^2^/s and IR-assisted RW™: 4.2 × 10^−10^ ≤ *Deff* ≤ 13.4 × 10^−10^ m^2^/s) and kinetic parameter values showed dependence on the drying temperature (60–90 °C), giving as a result of activation energies between 14.59 and 34.44 kJ/mol. When comparing experimental data to those predicted by the Midilli–Kucuk model, this model could be a useful tool for describing drying kinetics and estimating drying time under different RW™ drying method conditions. Furthermore, the IR-assisted system decreased RW™ drying process time by around 60% to reach an equilibrium moisture content of 10%. Therefore, combined treatments of IR and RW™ would be a good methodology to preserve dried *Physalis* fruit purée with a minimum impact on its quality properties, as well as reducing energy costs and consumption during the industrial-scale drying process.

## Figures and Tables

**Figure 1 foods-09-00343-f001:**
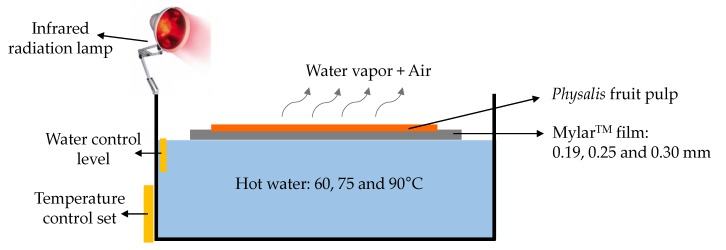
Experimental setup for infrared (IR)-assisted Refractance Window™ (RW™) drying experiments.

**Figure 2 foods-09-00343-f002:**
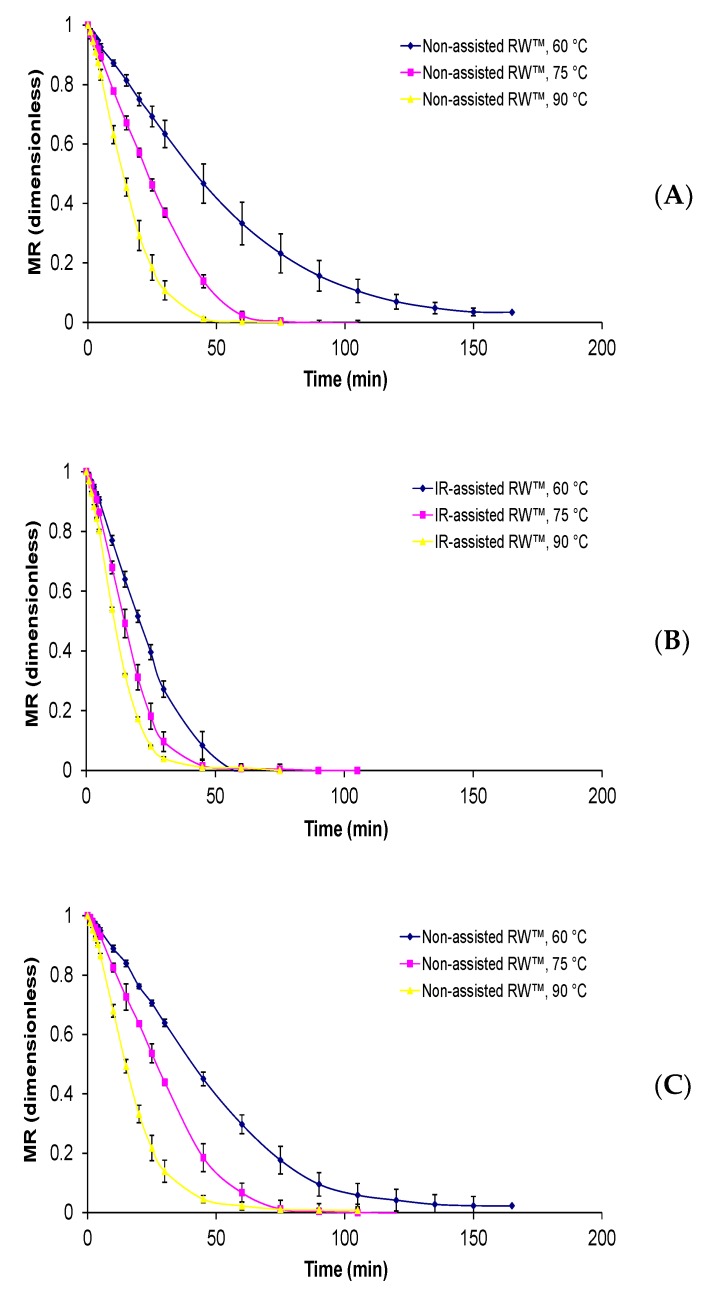
*Physalis* purée drying curves subjected to different RW™ drying conditions, where (**A**) non-assisted RW™ and (**B**) IR-assisted RW™ with Mylar™ thickness of 0.19 mm; (**C**) non-assisted RW™ and (**D**) IR-assisted RW™ with Mylar™ thickness of 0.25 mm; (**E**) non-assisted RW™ and (**F**) IR-assisted RW™ with Mylar™ thickness of 0.29 mm. Moisture ratio (MR).

**Figure 3 foods-09-00343-f003:**
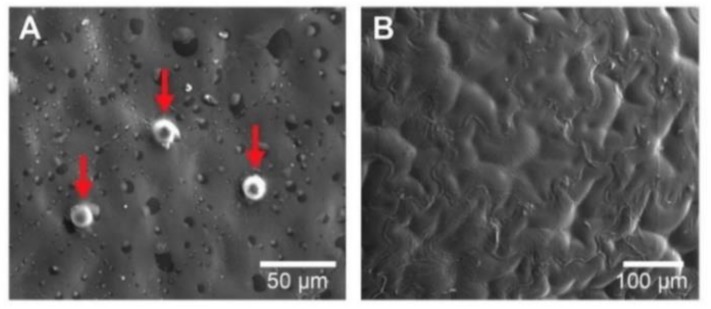
Microphotographs of peel and seed pieces from fresh *Physalis* purée of via SEM. (**A**) External surface of the skin pieces, red arrows indicate eminences in the exocarp of the fruit (500×). (**B**) External surface of the seed pieces (200×).

**Figure 4 foods-09-00343-f004:**
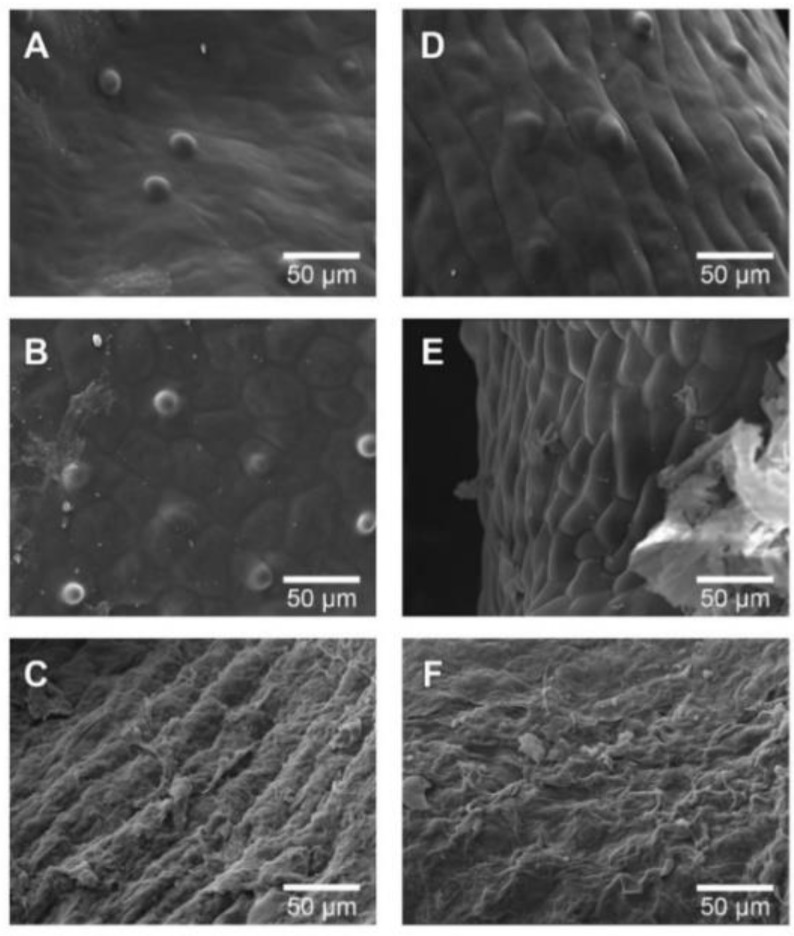
Microstructure of pieces of dried *Physalis* purée peel via SEM (500×). Non-assisted RW™ method at 60 °C (**A**), 75 °C (**B**), and 90 °C (**C**). IR-assisted RW™ method 60 °C (**D**), 75 °C (**E**), and 90 °C (**F**).

**Figure 5 foods-09-00343-f005:**
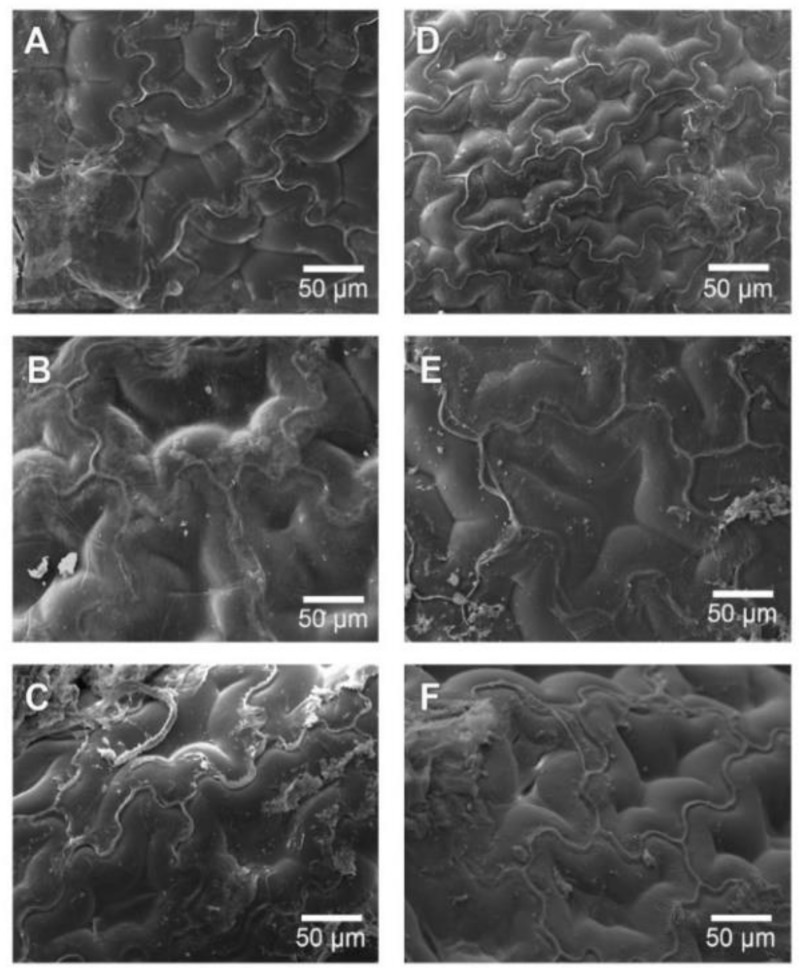
Microstructure of the dried *Physalis* purée seeds via SEM (200×). Non-assisted RW™ method at 60 °C (**A**), 75 °C (**B**), and 90 °C (**C**). IR-assisted RW™ method 60 °C (**D**), 75 °C (**E**), and 90 °C (**F**).

**Figure 6 foods-09-00343-f006:**
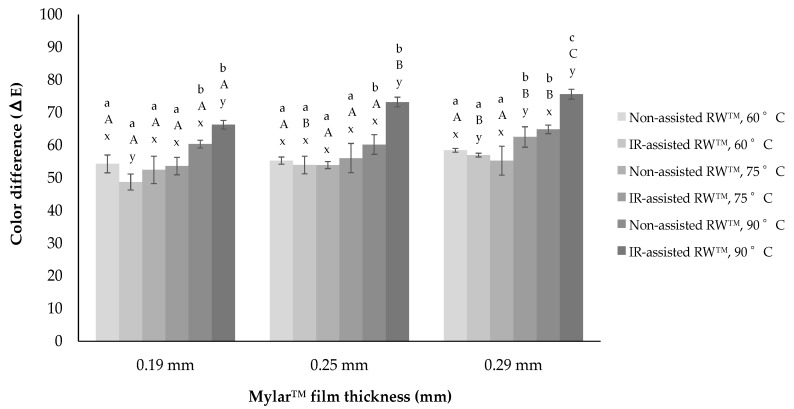
Color difference (*Δ*E) for dried P*hysalis* purée regarding to all RW™ drying experiments. Different lower case letters (a–c) indicate that there are significant differences (*p*-value < 0.05) between drying temperatures. Different upper case letters (A–C) indicate that there are significant differences (*p*-value < 0.05) between thickness. Different lower case letters (x, y, z) indicate that there are significant differences (*p*-value < 0.05) between RW™ drying method.

**Table 1 foods-09-00343-t001:** Thin-layer drying mathematical models.

N°	Equation	Model
1	*MR* = exp (−*k*t^n^)	Page
2	*MR* = *a*·exp(−*k*t)	Henderson–Pabis (HP)
3	*MR* = *a*·exp (−*k*t) + *b*·exp (−*k_o_*t) + *c*·exp (−*k*_1_t)	Modified Henderson–Pabis (M.HP)
4	MR = *a*·exp(−*k*_o_t) + *b*·exp(−*k*_1_t)	Two Term
5	MR = *a*·exp (−*k*t*^n^*) + *b*t	Midilli–Kucuk

*k*, *k*_o_ and *k*_1_ are kinetic parameters of each mathematical model; *n*, *a*, *b,* and *c* are empirical parameters for each mathematical model.

**Table 2 foods-09-00343-t002:** Moisture effective diffusivities (*D_eff_* × 10^−10^ m^2^/s) of dried *Physalis* purée subjected to different RW™ drying conditions.

T (°C)	Non-Assisted RW^TM^			IR-Assisted RW^TM^		
	Mylar^TM^ Thickness (mm)			Mylar^TM^ Thickness (mm)		
	0.19	0.25	0.29	0.19	0.25	0.29
60	2.74 ± 0.09 ^aAx^	3.34 ± 0.05 ^aBx^	3.95 ± 0.05 ^aBx^	4.26 ± 0.02 ^aAy^	4.20 ± 0.02 ^aAy^	6.08 ± 0.03 ^aBy^
75	4.26 ± 0.04 ^bAx^	4.56 ± 0.06 ^bAx^	5.78 ± 0.06 ^bBx^	8.21 ± 0.03 ^bAy^	10.30 ± 0.05 ^bBy^	10.38 ± 0.01 ^bBy^
90	9.42 ± 0.05 ^cABx^	9.12 ± 0.09 ^cBx^	10.07 ± 0.04 ^cAx^	11.93 ± 0.09 ^cAy^	11.24 ± 0.06 ^cAy^	13.44 ± 0.01 ^cBy^

Different lower case letters (a, b, c) in the same column indicate that there are significant differences (*p*-value < 0.05) between drying temperatures. Different upper case letters (A, B, C) in the same row indicate that there are significant differences (*p*-value < 0.05) between Mylar™ thickness. Different lower case letters (x, y, z) in the same row indicate that there are significant differences (*p*-value < 0.05) between RW™ drying method.

**Table 3 foods-09-00343-t003:** Kinetic and empirical parameters of each model to different non-assisted RW™ drying conditions.

Model	Parameter	Temperature (°C)—Mylar^TM^ Thickness (mm)								
		60—0.19			60—0.25			60—0.29		
HP	*a*	0.1	±	1.8 × 10^−3^	1.1	±	1.3 × 10^−2^	1.1	±	5.1 × 10^−3^
	*k*	2.0 × 10^−4^	±	1.4 × 10^−5^	2.1 × 10^−4^	±	6.7 × 10^−6^	2.1 × 10^−4^	±	3.0 × 10^−6^
M.HP	*a*	3.6 × 10^−1^	±	2.5 × 10^−3^	3.7 × 10^−1^	±	5.8 × 10^−3^	3.7 × 10^−1^	±	1.8 × 10^−3^
	*k*	2.0 × 10^−4^	±	2.1 × 10^−5^	2.1 × 10^−4^	±	6.7 × 10^−6^	2.1 × 10^−4^	±	3.0 × 10^−6^
	*b*	3.6 × 10^−1^	±	2.0 × 10^−3^	3.6 × 10^−1^	±	4.4 × 10^−3^	3.6 × 10^−1^	±	1.6 × 10^−3^
	*k_o_*	2.0 × 10^−4^	±	2.1 × 10^−5^	2.1 × 10^−4^	±	6.7 × 10^−6^	2.1 × 10^−4^	±	3.0 × 10^−6^
	*c*	3.5 × 10^−1^	±	1.3 × 10^−3^	3.6 × 10^−1^	±	3.2 × 10^−3^	3.6 × 10^−1^	±	1.8 × 10^−3^
	*k_1_*	2.0 × 10^−4^	±	2.1 × 10^−5^	2.1 × 10^−4^	±	6.7 × 10^−6^	2.1 × 10^−4^	±	3.0 × 10^−6^
Page	*k*	1.8 × 10^−8^	±	1.4 × 10^−8^	2.5 × 10^−9^	±	2.1 × 10^−9^	2.7 × 10^−9^	±	2.8 × 10^−9^
	*n*	2.1	±	9.8 × 10^−2^	2.4	±	2.0 × 10^−1^	2.5	±	3.3 × 10^−1^
Two Term	*a*	5.4 × 10^−1^	±	3.7 × 10^−3^	5.5 × 10^−1^	±	8.6 × 10^−3^	5.6 × 10^−1^	±	2.6 × 10^−3^
	*k_o_*	2.0 × 10^−4^	±	2.1 × 10^−5^	2.1 × 10^−4^	±	6.7 × 10^−6^	2.1 × 10^−4^	±	3.0 × 10^−6^
	*b*	5.3 × 10^−1^	±	2.1 × 10^−3^	5.4 × 10^−1^	±	4.8 × 10^−3^	5.4 × 10^−1^	±	2.6 × 10^−3^
	*k_1_*	2.0 × 10^−4^	±	2.1 × 10^−5^	2.1 × 10^−4^	±	6.7 × 10^−6^	2.1 × 10^−4^	±	3.0 × 10^−6^
Midilli–Kucuk	*a*	9.9 × 10^−1^	±	2.3 × 10^−3^	9.9 × 10^−1^	±	3.8 × 10^−3^	9.9 × 10^−1^	±	1.8 × 10^−3^
	*k*	2.5 × 10^−4^	±	1.8 × 10^−8^	2.6 × 10^−9^	±	2.2 × 10^−9^	2.5 × 10^−9^	±	3.0 × 10^−9^
	*b*	2.6 × 10^−5^	±	1.5 × 10^−5^	2.6 × 10^−5^	±	1.5 × 10^−5^	2.6 × 10^−5^	±	1.5 × 10^−5^
	*n*	2.0	±	9.6 × 10^−2^	2.4	±	2.1 × 10^−1^	2.5	±	3.6 × 10^−1^
		**75—0.19**			**75—0.25**			**75—0.29**		
HP	*a*	1.1	±	3.3 × 10^−3^	1.1	±	9.3 × 10^−3^	1.1	±	6.8 × 10^−4^
	*k*	3.3 × 10^−4^	±	6.0 × 10^−6^	2.6 × 10^−4^	±	2.7 × 10^−5^	2.9 × 10^−4^	±	1.2 × 10^−5^
M.HP	*a*	3.7 × 10^−1^	±	1.3 × 10^−3^	3.7 × 10^−1^	±	4.7 × 10^−3^	3.7 × 10^−1^	±	2.0 × 10^−3^
	*k*	3.3 × 10^−4^	±	6.0 × 10^−6^	2.6 × 10^−4^	±	2.7 × 10^−5^	2.9 × 10^−4^	±	4.5 × 10^−6^
	*b*	3.6 × 10^−1^	±	1.1 × 10^−3^	3.6 × 10^−1^	±	2.5 × 10^−3^	3.6 × 10^−1^	±	2.1 × 10^−3^
	*k_o_*	3.3 × 10^−4^	±	6.0 × 10^−6^	2.6 × 10^−4^	±	2.7 × 10^−5^	2.9 × 10^−4^	±	4.5 × 10^−6^
	*c*	3.6 × 10^−1^	±	9.6 × 10^−4^	3.6 × 10^−1^	±	2.2 × 10^−3^	3.6 × 10^−1^	±	2.1 × 10^−3^
	*k_1_*	3.3 × 10^−4^	±	6.0 × 10^−6^	2.6 × 10^−4^	±	2.7 × 10^−5^	2.9 × 10^−4^	±	4.5 × 10^−6^
Page	*k*	1.2 × 10^−9^	±	1.3 × 10^−9^	1.7 × 10^−7^	±	3.0 × 10^−7^	2.7 × 10^−9^	±	2.6 × 10^−9^
	*n*	2.6	±	1.5 × 10^−1^	2.4	±	6.1 × 10^−1^	2.5	±	1.5 × 10^−1^
Two Term	*a*	5.5 × 10^−1^	±	7.2 × 10^−3^	5.5 × 10^−1^	±	6.5 × 10^−3^	5.5 × 10^−1^	±	2.9 × 10^−3^
	*k_o_*	3.1 × 10^−4^	±	3.1 × 10^−5^	2.6 × 10^−4^	±	2.7 × 10^−5^	2.9 × 10^−4^	±	4.4 × 10^−6^
	*b*	5.4 × 10^−1^	±	2.1 × 10^−3^	5.4 × 10^−1^	±	3.1 × 10^−3^	5.4 × 10^−1^	±	2.5 × 10^−3^
	*k_1_*	3.1 × 10^−4^	±	3.1 × 10^−5^	2.6 × 10^−4^	±	2.7 × 10^−5^	2.9 × 10^−4^	±	4.4 × 10^−6^
Midilli–Kucuk	*a*	9.8 × 10^−1^	±	3.5 × 10^−3^	1.0	±	5.4 × 10^−2^	9.9 × 10^−1^	±	3.9 × 10^−3^
	*k*	6.7 × 10^−10^	±	7.2 × 10^−10^	2.6 × 10^−9^	±	4.1 × 10^−9^	2.3 × 10^−9^	±	2.5 × 10^−9^
	*b*	2.6 × 10^−5^	±	1.5 × 10^−5^	2.6 × 10^−5^	±	1.5 × 10^−5^	2.6 × 10^−5^	±	1.5 × 10^−5^
	*n*	2.7	±	1.8 × 10^−1^	1.7	±	1.5	2.5	±	1.8 × 10^−1^
		**90—0.19**			**90—0.25**			**90—0.29**		
HP	*a*	1.1	±	5.6 × 10^−3^	1.1	±	6.2 × 10^−3^	1.1	±	4.2 × 10^−3^
	*k*	5.5 × 10^−4^	±	6.1 × 10^−5^	5.1 × 10^−4^	±	3.5 × 10^−5^	5.2 × 10^−4^	±	2.3 × 10^−5^
M.HP	*a*	3.7 × 10^−1^	±	2.7 × 10^−3^	3.7 × 10^−1^	±	2.4 × 10^−3^	3.7 × 10^−1^	±	1.6 × 10^−3^
	*k*	5.5 × 10^−4^	±	6.1 × 10^−5^	5.1 × 10^−4^	±	3.5 × 10^−5^	5.2 × 10^−4^	±	2.3 × 10^−5^
	*b*	3.6 × 10^−1^	±	1.8 × 10^−3^	3.7 × 10^−1^	±	2.1 × 10^−3^	3.7 × 10^−1^	±	1.4 × 10^−3^
	*k_o_*	5.5 × 10^−4^	±	6.1 × 10^−5^	5.1 × 10^−4^	±	3.5 × 10^−5^	5.2 × 10^−4^	±	2.3 × 10^−5^
	*c*	3.6 × 10^−1^	±	1.1 × 10^−3^	3.6 × 10^−1^	±	1.8 × 10^−3^	3.6 × 10^−1^	±	1.5 × 10^−3^
	*k_1_*	5.5 × 10^−4^	±	6.1 × 10^−5^	5.1 × 10^−4^	±	3.5 × 10^−5^	5.2 × 10^−4^	±	2.3 × 10^−5^
Page	*k*	5.1 × 10^−8^	±	2.6 × 10^−8^	3.3 × 10^−7^	±	2.1 × 10^−7^	2.0 × 10^−7^	±	1.0 × 10^−7^
	*n*	2.3	±	6.6 × 10^−2^	2.0	±	1.4 × 10^−1^	2.0	±	5.9 × 10^−2^
Two Term	*a*	5.5 × 10^−1^	±	4.0 × 10^−3^	5.5 × 10^−1^	±	3.5 × 10^−3^	5.5 × 10^−1^	±	2.3 × 10^−3^
	*k_o_*	5.5 × 10^−4^	±	6.1 × 10^−5^	5.1 × 10^−4^	±	3.5 × 10^−5^	5.2 × 10^−4^	±	2.3 × 10^−5^
	*b*	5.4 × 10^−1^	±	1.6 × 10^−3^	5.4 × 10^−1^	±	2.7 × 10^−3^	5.4 × 10^−1^	±	2.3 × 10^−3^
	*k_1_*	5.5 × 10^−4^	±	6.1 × 10^−5^	5.1 × 10^−4^	±	3.5 × 10^−5^	5.2 × 10^−4^	±	2.3 × 10^−5^
Midilli–Kucuk	*a*	9.9 × 10^−1^	±	5.2 × 10^−3^	9.9 × 10^−1^	±	1.3 × 10^−3^	9.9 × 10^−1^	±	2.9 × 10^−3^
	*k*	6.8 × 10^−8^	±	3.8 × 10^−8^	1.8 × 10^−7^	±	1.3 × 10^−7^	1.3 × 10^−7^	±	1.1 × 10^−7^
	*b*	2.6 × 10^−5^	±	1.5 × 10^−5^	1.5 × 10^−5^	±	1.7 × 10^−5^	1.6 × 10^−5^	±	1.7 × 10^−5^
	*n*	2.2	±	1.0 × 10^−1^	2.1	±	1.3 × 10^−1^	2.1	±	1.6 × 10^−1^

**Table 4 foods-09-00343-t004:** Kinetic and empirical parameters of each model to different IR-assisted RW™ drying conditions.

Model	Parameter	Temperature (°C)—Mylar^TM^ Thickness (mm)								
		60—0.19			60—0.25			60—0.29		
HP	*a*	1.1	±	8.5 × 10^−3^	1.1	±	4.7 × 10^−3^	1.1	±	9.3 × 10^−4^
	*k*	3.3 × 10^−4^	±	2.0 × 10^−5^	3.4 × 10^−4^	±	3.5 × 10^−5^	3.2 × 10^−4^	±	3.8 × 10^−6^
M.HP	*a*	3.6 × 10^−1^	±	4.2 × 10^−3^	3.6 × 10^−1^	±	2.4 × 10^−3^	3.6 × 10^−1^	±	4.5 × 10^−4^
	*k*	3.3 × 10^−4^	±	2.0 × 10^−5^	3.4 × 10^−4^	±	3.5 × 10^−5^	3.2 × 10^−4^	±	3.8 × 10^−6^
	*b*	3.6 × 10^−1^	±	2.6 × 10^−3^	3.6 × 10^−1^	±	1.5 × 10^−3^	3.6 × 10^−1^	±	2.6 × 10^−4^
	*k_o_*	3.3 × 10^−4^	±	2.0 × 10^−5^	3.4 × 10^−4^	±	3.5 × 10^−5^	3.2 × 10^−4^	±	3.8 × 10^−6^
	*c*	3.6 × 10^−1^	±	2.1 × 10^−3^	3.6 × 10^−1^	±	8.7 × 10^−4^	3.6 × 10^−1^	±	2.3 × 10^−4^
	*k_1_*	3.3 × 10^−4^	±	2.0 × 10^−5^	3.4 × 10^−4^	±	3.5 × 10^−5^	3.2 × 10^−4^	±	3.8 × 10^−6^
Page	*k*	1.3 × 10^−9^	±	9.3 × 10^−10^	3.8 × 10^−10^	±	4.6 × 10^−10^	2.1 × 10^−10^	±	1.5 × 10^−10^
	*n*	2.7	±	1.4 × 10^−1^	2.8	±	1.5 × 10^−1^	2.9	±	8.4 × 10^−2^
Two Term	*a*	5.4 × 10^−1^	±	6.1 × 10^−3^	5.5 × 10^−1^	±	3.5 × 10^−3^	5.5 × 10^−1^	±	6.0 × 10^−4^
	*k_o_*	3.3 × 10^−4^	±	2.0 × 10^−5^	3.4 × 10^−4^	±	3.5 × 10^−5^	3.2 × 10^−4^	±	3.8 × 10^−6^
	*b*	5.3 × 10^−1^	±	3.2 × 10^−3^	5.4 × 10^−1^	±	1.3 × 10^−3^	5.4 × 10^−1^	±	2.9 × 10^−4^
	*k_1_*	3.3 × 10^−4^	±	2.0 × 10^−5^	3.4 × 10^−4^	±	3.5 × 10^−5^	3.2 × 10^−5^	±	3.8 × 10^−6^
Midilli–Kucuk	*a*	1.0	±	3.2 × 10^−2^	1.0	±	4.4 × 10^−2^	1.0	±	4.6 × 10^−2^
	*k*	3.1 × 10^−11^	±	3.0 × 10^−11^	3.1 × 10^−11^	±	2.7 × 10^−11^	3.9 × 10^−11^	±	3.7 × 10^−11^
	*b*	2.2 × 10^−4^	±	7.9 × 10^−5^	1.2 × 10^−4^	±	1.4 × 10^−4^	1.8 × 10^−4^	±	1.3 × 10^−4^
	*n*	1.4	±	2.5	2.2	±	1.4	9.7 × 10^−1^	±	1.7
		**75—0.19**			**75—0.25**			**75—0.29**		
HP	*a*	1.1	±	1.0 × 10^−4^	1.1	±	2.1 × 10^−3^	1.1	±	1.3 × 10^−3^
	*k*	5.3 × 10^−4^	±	1.1 × 10^−5^	5.1 × 10^−4^	±	1.5 × 10^−5^	5.3 × 10^−4^	±	3.6 × 10^−5^
M.HP	*a*	3.7 × 10^−1^	±	1.7 × 10^−4^	3.7 × 10^−1^	±	8.0 × 10^−4^	3.7 × 10^−1^	±	9.8 × 10^−4^
	*k*	5.3 × 10^−4^	±	1.1 × 10^−5^	5.1 × 10^−4^	±	1.5 × 10^−5^	5.3 × 10^−4^	±	3.6 × 10^−5^
	*b*	3.7 × 10^−1^	±	1.0 × 10^−4^	3.6 × 10^−1^	±	7.0 × 10^−4^	3.6 × 10^−1^	±	4.2 × 10^−4^
	*k_o_*	5.3 × 10^−4^	±	1.1 × 10^−5^	5.1 × 10^−4^	±	1.5 × 10^−5^	5.3 × 10^−4^	±	3.6 × 10^−5^
	*c*	3.6 × 10^−1^	±	5.8 × 10^−5^	3.6 × 10^−1^	±	6.1 × 10^−4^	3.6 × 10^−1^	±	3.2 × 10^−4^
	*k_1_*	5.3 × 10^−4^	±	1.1 × 10^−5^	5.1 × 10^−4^	±	1.5 × 10^−5^	5.3 × 10^−4^	±	3.6 × 10^−5^
Page	*k*	3.2 × 10^−9^	±	1.6 × 10^−9^	1.4 × 10^−9^	±	6.8 × 10^−10^	1.1 × 10^−8^	±	7.3 × 10^−9^
	*n*	2.6	±	7.4 × 10^−2^	2.7	±	6.2 × 10^−2^	2.5	±	1.2 × 10^−1^
Two Term	*a*	5.5 × 10^−1^	±	2.6 × 10^−4^	5.5 × 10^−1^	±	1.2 × 10^−3^	5.5 × 10^−1^	±	1.4 × 10^−3^
	*k_o_*	5.3 × 10^−4^	±	1.1 × 10^−5^	5.1 × 10^−4^	±	1.5 × 10^−5^	5.3 × 10^−4^	±	3.6 × 10^−5^
	*b*	5.4 × 10^−1^	±	1.7 × 10^−4^	5.4 × 10^−1^	±	9.5 × 10^−4^	5.4 × 10^−1^	±	4.7 × 10^−4^
	*k_1_*	5.3 × 10^−4^	±	1.1 × 10^−5^	5.1 × 10^−4^	±	1.5 × 10^−5^	5.3 × 10^−4^	±	3.6 × 10^−5^
Midilli–Kucuk	*a*	1.0	±	3.5 × 10^−2^	1.0	±	4.8 × 10^−2^	9.9 × 10^−1^	±	2.3 × 10^−3^
	*k*	1.9 × 10^−9^	±	1.9 × 10^−9^	9.0 × 10^−10^	±	8.9 × 10^−10^	1.8 × 10^−8^	±	9.0 × 10^−9^
	*b*	1.1 × 10^−4^	±	1.8 × 10^−4^	1.3 × 10^−4^	±	2.0 × 10^−4^	2.4 × 10^−5^	±	9.1 × 10^−6^
	*n*	3.3	±	1.2	1.8	±	1.6	2.4	±	8.4 × 10^−2^
		**90—0.19**			**90—0.25**			**90—0.29**		
HP	*a*	1.1	±	2.9 × 10^−3^	1.1	±	2.1 × 10^−3^	1.1	±	5.6 × 10^−3^
	*k*	8.0 × 10^−4^	±	3.4 × 10^−5^	7.8 × 10^−4^	±	3.4 × 10^−5^	8.5 × 10^−4^	±	5.2 × 10^−5^
M.HP	*a*	3.8 × 10^−1^	±	1.6 × 10^−3^	3.7 × 10^−1^	±	6.5 × 10^−4^	3.8 × 10^−1^	±	2.9 × 10^−3^
	*k*	8.0 × 10^−4^	±	3.4 × 10^−5^	7.8 × 10^−4^	±	3.4 × 10^−5^	8.5 × 10^−4^	±	5.2 × 10^−5^
	*b*	3.7 × 10^−1^	±	8.9 × 10^−4^	3.7 × 10^−1^	±	7.1 × 10^−4^	3.7 × 10^−1^	±	1.8 × 10^−3^
	*k_o_*	8.0 × 10^−4^	±	3.4 × 10^−5^	7.8 × 10^−4^	±	3.4 × 10^−5^	8.5 × 10^−4^	±	5.2 × 10^−5^
	*c*	3.6 × 10^−1^	±	4.0 × 10^−4^	3.6 × 10^−1^	±	9.6 × 10^−4^	3.6 × 10^−1^	±	1.1 × 10^−3^
	*k_1_*	8.0 × 10^−4^	±	3.4 × 10^−5^	7.8 × 10^−4^	±	3.4 × 10^−5^	8.5 × 10^−4^	±	5.2 × 10^−5^
Page	*k*	2.8 × 10^−8^	±	2.4 × 10^−9^	5.3 × 10^−8^	±	3.1 × 10^−8^	1.7 × 10^−8^	±	7.4 × 10^−9^
	*n*	2.4	±	2.0 × 10^−2^	2.4	±	6.7 × 10^−2^	2.5	±	8.5 × 10^−2^
Two Term	*a*	5.6 × 10^−1^	±	2.3 × 10^−3^	5.6 × 10^−1^	±	9.5 × 10^−4^	5.7 × 10^−1^	±	4.2 × 10^−3^
	*k_o_*	8.0 × 10^−4^	±	3.4 × 10^−5^	7.8 × 10^−4^	±	3.4 × 10^−5^	8.5 × 10^−4^	±	5.2 × 10^−5^
	*b*	5.4 × 10^−1^	±	6.6 × 10^−4^	5.4 × 10^−1^	±	1.5 × 10^−3^	5.4 × 10^−1^	±	1.6 × 10^−3^
	*k_1_*	8.0 × 10^−4^	±	3.4 × 10^−5^	7.8 × 10^−4^	±	3.3 × 10^−5^	8.5 × 10^−4^	±	5.2 × 10^−5^
Midilli–Kucuk	*a*	9.9 × 10^−1^	±	4.7 × 10^−3^	1.0	±	1.2 × 10^−3^	9.9 × 10^−1^	±	5.1 × 10^−3^
	*k*	3.4 × 10^−8^	±	2.7 × 10^−9^	7.1 × 10^−8^	±	4.5 × 10^−8^	2.1 × 10^−8^	±	1.4 × 10^−8^
	*b*	7.1 × 10^−5^	±	2.5 × 10^−5^	9.7 × 10^−5^	±	2.0 × 10^−5^	4.7 × 10^−5^	±	2.7 × 10^−5^
	*n*	2.4	±	3.4 × 10^−2^	2.3	±	8.6 × 10^−2^	2.5	±	1.7 × 10^−1^

**Table 5 foods-09-00343-t005:** Changes in *L** coordinate values of dried *Physalis* purée subjected to different RW™ drying conditions.

T (°C)	Non-Assisted RW™			IR-Assisted RW™		
	Mylar™ Thickness (mm)			Mylar™ Thickness (mm)		
	0.19	0.25	0.29	0.19	0.25	0.29
60	46.02 ± 3.96 ^bAx^	44.47 ± 3.16 ^aAx^	40.60 ± 1.90 ^aBx^	47.43 ± 4.59 ^aAx^	43.99 ± 5.65 ^aBx^	42.84 ± 5.21 ^aBx^
75	45.08 ± 4.11 ^abAx^	47.99±1.98 ^aAx^	43.80 ± 0.66 ^bBx^	44.44 ± 2.97 ^aAx^	39.99 ± 3.79 ^aBy^	38.43 ± 5.28 ^aBy^
90	41.41 ± 2.40 ^aBx^	35.84±5.71 ^bABx^	38.63 ± 0.73 ^aAx^	36.42 ± 3.99 ^bAy^	31.66 ± 3.32 ^bBx^	29.95 ± 5.25 ^bBy^

Different lower case letters (a, b, c) in the same column indicate that there are significant differences (*p*-value < 0.05) between drying temperatures. Different upper case letters (A, B, C) in the same row indicate that there are significant differences (*p*-value < 0.05) between Mylar™ thickness. Different lower case letters (x, y, z) in the same row indicate that there are significant differences (*p*-value < 0.05) between RW™ drying method.

**Table 6 foods-09-00343-t006:** Changes in *a** coordinate values of dried *Physalis* purée subjected to different RW™ drying conditions.

T (°C)	Non-Assisted RW™			IR-Assisted RW™		
	Mylar™ Thickness (mm)			Mylar™ Thickness (mm)		
	0.19	0.25	0.29	0.19	0.25	0.29
60	18.40 ± 3.58 ^aAx^	19.66 ± 4.17 ^aAx^	24.83 ± 3.47 ^aBx^	22.39 ± 3.37 ^aAy^	21.73 ± 2.08 ^aAx^	21.58 ± 4.78 ^aAy^
75	22.54 ± 2.10 ^aAx^	20.25 ± 2.55 ^aAx^	21.76 ± 4.31 ^aAx^	22.27 ± 3.98 ^aABx^	25.05 ± 3.53 ^bAy^	21.86 ± 3.84 ^aBx^
90	21.35 ± 2.86 ^aAx^	18.50 ± 1.90 ^aBx^	21.59 ± 2.64 ^aAx^	22.23 ± 0.96 ^aAx^	20.47 ± 4.42 ^aAx^	17.52 ± 1.16 ^bBy^

Different lower case letters (a, b, c) in the same column indicate that there are significant differences (*p*-value < 0.05) between drying temperatures. Different upper case letters (A, B, C) in the same row indicate that there are significant differences (*p*-value < 0.05) between Mylar™ thickness. Different lower case letters (x, y, z) in the same row indicate that there are significant differences (*p*-value < 0.05) between RW™ drying method.

**Table 7 foods-09-00343-t007:** Changes in *b** coordinate values of dried *Physalis* purée subjected to different RW™ drying conditions.

T (°C)	Non-Assisted RW™			IR-Assisted RW^v^		
	Mylar™ Thickness (mm)			Mylar™ Thickness (mm)		
	0.19	0.25	0.29	0.19	0.25	0.29
60	37.12 ± 2.18 ^aAx^	36.69 ± 2.60 ^cAx^	34.47 ± 0.96 ^aBx^	42.55 ± 5.16 ^aAy^	38.57 ± 1.56 ^aBx^	35.16 ± 4.67 ^aBx^
75	39.32 ± 5.78 ^aABx^	42.54 ± 2.50 ^bAx^	36.68 ± 3.78 ^aBx^	38.30 ± 2.99 ^aAx^	34.14 ± 3.72 ^aBy^	31.47 ± 1.43 ^abBy^
90	31.72 ± 1.20 ^bAx^	32.64 ± 7.08 ^aAx^	27.92 ± 1.21 ^bBx^	27.90 ± 3.43 ^bAy^	23.13 ± 5.63 ^bBy^	22.32 ± 1.41 ^bBy^

Different lower case letters (a, b, c) in the same column indicate that there are significant differences (*p*-value < 0.05) between drying temperatures. Different upper case letters (A, B, C) in the same row indicate that there are significant differences (*p*-value < 0.05) between Mylar™ thickness. Different lower case letters (x, y, z) in the same row indicate that there are significant differences (*p*-value < 0.05) between RW™ drying method.

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
