# Peer review of "Effects of Infrared-Assisted Refractance Window™ Drying on the Drying Kinetics, Microstructure, and Color of Physalis Fruit Purée"

_foods, 2020, doi:10.3390/foods9030343_

Round 1

Reviewer 1 Report

The manuscript was improved.

Author Response

Comments and Suggestions for Authors

Reviewer 1: The manuscript was improved.

Authors: We want to thanks the reviewer’s comment.

Reviewer 2 Report

The article presents important information on drying Physalis fruit using alternative methods (RW™ method).  The interest in research on Physalis is very important due to the bioactive properties of this fruit.

The research can certainly reduce the cost of the drying process without any physicochemical or organoleptic losses of raw material.

In this work, the experience was well planned and a graphical diagram of the experiment was presented.

Mathematical models have also been developed using many equations well describing the effect of the drying phenomenon.

However, there is a lack of information whether these techniques can be applied on an industrial scale, which is worth mentioning.

Generally, the work was written very carefully, the results were very well developed statistically and mathematically.

In my opinion, the paper is suitable for publication in the Foods journal after a minor addition.

Author Response

Comments and Suggestions for Authors

Reviewer 2: The article presents important information on drying Physalis fruit using alternative methods (RW™ method).  The interest in research on Physalis is very important due to the bioactive properties of this fruit.

Authors: We want to thanks the reviewer’s comments.

Reviewer 2: The research can certainly reduce the cost of the drying process without any physicochemical or organoleptic losses of raw material.

Authors: We want to thanks the reviewer’s comments.

Reviewer 2: In this work, the experience was well planned and a graphical diagram of the experiment was presented.

Authors: We want to thanks the reviewer’s comments.

Reviewer 2: Mathematical models have also been developed using many equations well describing the effect of the drying phenomenon.

Authors: We want to thanks the reviewer’s comments.

Reviewer 2: However, there is a lack of information whether these techniques can be applied on an industrial scale, which is worth mentioning.

Authors: We have included new sentences and improved the text of manuscript about industrial importance. Also, this month, we have received the pilot-scale RW™ dryer equipment which we hope to work issues related to industrial-escalation process, heat flows, energy consumption and costs, among others. 

Reviewer 2: Generally, the work was written very carefully, the results were very well developed statistically and mathematically.

Authors: We want to thanks the reviewer’s comments.

Reviewer 2: In my opinion, the paper is suitable for publication in the Foods journal after a minor addition.

Authors: We have improved the manuscript about the comment’s reviewer.

Reviewer 3 Report

Manuscript: foods-730963-peer-review-v1

Authors: Luis Puente-Díaz, Oliver Spolmann, Diego Nocetti, Liliana Zura-Bravo, and Roberto Lemus-Mondaca 

Titlle: Effects of infrared-assisted Refractance Window™ drying on the drying kinetics, microstructure and color of Physalis fruit purée 

Line 47: freeze-dryer.

Line 50: saqme or same?

Lines 55-61: Can you specify the frequency/wavelength that IR drying works?

Line 67: structural or macrostructure? Structural involves macro- and micro-structure.

Line 88-95: Did you determine the maturity index of the fruits used in the experiment? How the fruits were sanitized and/or conditioned prior to obtaining the purée. How did you prevent the enzymatic browning?

Line 93: Why did you packaged the purée in polyethylene bags? How long the purée remained packaged until drying? Can you provide the storage temperature?  The packaging was performed under vacuum conditions? It should be taking into account that the storage of the purée, temperature, time and the conditions of the process dramatically influence in the quality of the product, specially in texture and color that you consider to be the most important quality parameters (line 62).

Line 93-95: Can you provide the temperature and time used to determine the moisture? Did you measure sugar content of the purée? And water activity?

Other important parameter to analyze is the polyphenol and carotenoid content and the antioxidant activity due to the high content that you mentioned in line 66.

Line 97-108: Can you provide more details regarding to the IR radiation lamp? Provider, frequency, wavelength, etc?

Did you measure the temperature of the puree??

The weight changes were measured only at before and after drying process? Did you measure during the drying process? Can you explain how do you take off the mylar film from the purée? Because if this separation is not appropriate it could be affect the final weigth.

Why did you choose 60, 75 and 90 ºC? from my point of view these temperatures are very high. Firstly, you are applying a coupled drying (HAD + IR), thus the selected temperatures should be lower than only HAD. In addition, over 60 ºC, sugar browning phenomena occurs.

Line 116: to use?

Line 126: Can you better explain the consideration of Me equal to 0? Why you affirm that is very small comparing to Mt and M0.

Line 165: Please provide manufacture, state and country of the colorimeter.

Line 201: A space is missing after the dot.

Figure 2: Which parameter is MR? it is not explained in the text or in the caption.

It should be important to introduce a nomenclature section.

Author Response

Comments and Suggestions for Authors

Manuscript: foods-730963-peer-review-v1

Authors: Luis Puente-Díaz, Oliver Spolmann, Diego Nocetti, Liliana Zura-Bravo, and Roberto Lemus-Mondaca 

Title: Effects of infrared-assisted Refractance Window™ drying on the drying kinetics, microstructure and color of Physalis fruit purée 

Reviewer 3: Line 47: freeze-dryer.

Authors: The word has been corrected.

Reviewer 3: Line 50: saqme or same?

Authors: The word has been corrected.

Reviewer 3: Lines 55-61: Can you specify the frequency/wavelength that IR drying works?

Authors: We have improved the sentence.

Reviewer 3: Line 67: structural or macrostructure? Structural involves macro- and micro-structure.

Authors: The sentence has been corrected.

Reviewer 3: Line 88-95: Did you determine the maturity index of the fruits used in the experiment? How the fruits were sanitized and/or conditioned prior to obtaining the purée. How did you prevent the enzymatic browning?

Authors: We have improved the sentence. The maturity index of Physalis fruits was just a visual measurement that taking account of mechanical damage, cuts, perforations, and impacts. The fruits just were handed-washed with tap water, then these were triturated to obtain the Physalis purée. In this case, a good cold chain (5 °C) was used, from producer Hortifrut Ltd. to our laboratory, in order to prevent enzymatic browning by reducing the rate of reaction.

Reviewer 3: Line 93: Why did you packaged the purée in polyethylene bags? How long the purée remained packaged until drying? Can you provide the storage temperature?  The packaging was performed under vacuum conditions? It should be taking into account that the storage of the purée, temperature, time and the conditions of the process dramatically influence in the quality of the product, especially in texture and color that you consider to be the most important quality parameters (line 62).

Authors: We used low-density polyethylene plastics because these are flexible with good elongation before breakage and good puncture resistance, as well as has a fair moisture barrier and oxygen barrier. Also, the low-density polyethylene has high clarity, is chemically inert, and has good impact strength and excellent tear and stress crack resistance. The samples frozen were keeping stored until 15 days for obtaining drying kinetics data, then samples dried were stored under refrigerated conditions (5 °C × 5 days) until microstructure and color analysis.

Reviewer 3: Line 93-95: Can you provide the temperature and time used to determine the moisture? Did you measure sugar content of the purée? And water activity?

Authors: The sentence has been improved. We have included temperature and time used to determine the moisture content of purée sample, as well as the measurement of sugar content and water activity (M&M and R&D).

Reviewer 3: Other important parameter to analyze is the polyphenol and carotenoid content and the antioxidant activity due to the high content that you mentioned in line 66.

Authors: We agree with the reviewer, however, in this first research stage we are evaluating physical features and drying kinetics, we hope to perform for the second stage the different research activities about polyphenol content, carotenoid content and the antioxidant activity and to evaluate as these properties will be affected by diverse RW™ drying conditions.

Reviewer 3: Line 97-108: Can you provide more details regarding to the IR radiation lamp? Provider, frequency, wavelength, etc?

Authors: The sentence has been corrected.

Reviewer 3: Did you measure the temperature of the puree??

Authors: Effectively, we had measured the temperature of Physalis purée, but only for 65 and 70 °C (non-assisted RW™ drying), and this way we could observe that the purée samples reach drying temperature quickly and then this was kept constant during the whole drying process. Thereby, we think that these results are interesting but they are not complete to be presented in the manuscript.

Reviewer 3: The weight changes were measured only at before and after drying process? Did you measure during the drying process? Can you explain how do you take off the mylar film from the purée? Because if this separation is not appropriate it could be affect the final weight.

Authors: The sentence has been corrected in the manuscript text.

“Three containers were built from Mylar™ film, these have an effective transfer area of 140 cm2 where Physalis purée was placed and then spread over the Mylar™ film aided by a spatula blade with a 3 mm gap”.

Reviewer 3: Why did you choose 60, 75 and 90 ºC? from my point of view these temperatures are very high. Firstly, you are applying a coupled drying (HAD + IR), thus the selected temperatures should be lower than only HAD. In addition, over 60 ºC, sugar browning phenomena occurs.

Authors: We agree to the reviewer about high temperatures used but in this case, the RW™ drying times are shorter than air-convective drying, in addition, the addition of IR heating caused an increase drying rate. As for sugar browning phenomena, we already have included a discussion about this issue based on chromatic coordinates (L*, a* and b*). In research subsequently, we are going to consider the measurement of non-enzymatic browning along with bioactive compounds changes. Also, we have reviewed the literature specialized about RW™ drying process, and there are researches that are working under a range temperature similar to ours.

  • Franco, S.; Jaques, A.; Pinto, M.; Fardella, M.; Valencia, P.; Núñez, H.; Ramírez, C.; Simpson, R.. Dehydration of salmon (Atlantic salmon), beef, and apple (Granny Smith) using Refractance window™: Effect on diffusion behavior, texture, and color changes. Inn Food Sci Emer Technol 2019, 52, 8-16.
  • Azizi, D.; Jafari, S.M.; Mirzaei, H.; Dehnad, D. The inflluence of refractance window drying on qualitative properties of kiwifruit slices. Int J Food Eng 2016, 13, 20160201.
  • Kaur, G.; Saha, S.; Kumari, K.; Datta, A.K. Mango pulp drying by Refractance Window method. Agr Eng Int: CIGR J 2017 19, 145-151.

Reviewer 3: Line 116: to use?

Authors: The word has been corrected.

Reviewer 3: Line 126: Can you better explain the consideration of Me equal to 0? Why you affirm that is very small comparing to Mt and M0.

Authors: we made a mistake when writing that sentence (Me ≅ 0) because of the equilibrium moisture content experimental data whether were included in the calculation of moisture ratio (MR) data.

Reviewer 3: Line 165: Please provide manufacture, state and country of the colorimeter.

Authors: The sentence has been corrected.

Reviewer 3: Line 201: A space is missing after the dot.

Authors: The sentence has been corrected.

Reviewer 3: Figure 2: Which parameter is MR? it is not explained in the text or in the caption.

It should be important to introduce a nomenclature section.

Authors: We have corrected the sentence regard to Moisture Ratio (MR, dimensionless).

Round 2

Reviewer 3 Report

Manuscript: foods-730963-peer-review-v2 Authors: Luis Puente-Díaz, Oliver Spolmann, Diego Nocetti, Liliana Zura-Bravo, and Roberto Lemus-Mondaca  Titlle: Effects of infrared-assisted Refractance Window™ drying on the drying kinetics, microstructure and color of Physalis fruit purée  Line 93-95: The visual inspection is not the maturity index of the fruits. For future works is important to measure and classify the fruits according to the MI. Answer line 93: Authors did not mention if the packaged was under vacuum conditions. I do not understand why the samples were frozen, even more considering that no pre-treatments were done to avoid enzymatic browning. After freezing the structure of the fruit tissue is destroyed due to the presence of the ice crystals. And I insist, especially if one of the main parameters measured was the color. Line 111: cm2 must be in superscript. The following part of the question is not answered: “The weight changes were measured only at before and after drying process? Did you measure during the drying process?. Lines 108-109: You mentioned that drying times are shorter than air-convective drying, however drying times are not described in the manuscript. Equation 2. Why did you work with ratios of moisture??

Author Response

Comments and Suggestions for Authors

Manuscript: foods-730963-peer-review-v2

Authors: Luis Puente-Díaz, Oliver Spolmann, Diego Nocetti, Liliana Zura-Bravo, and Roberto Lemus-Mondaca

Title: Effects of infrared-assisted Refractance Window™ drying on the drying kinetics, microstructure and color of Physalis fruit purée 

Reviewer: Line 93-95: The visual inspection is not the maturity index of the fruits. For future works is important to measure and classify the fruits according to the MI.

Authors: We agree with the reviewer about MI method used, and without a doubt, we will consider your recommendations in our future experiments/research.

Reviewer: Answer line 93: Authors did not mention if the packaged was under vacuum conditions.

Authors: We have corrected the sentence.

Reviewer: I do not understand why the samples were frozen, even more considering that no pre-treatments were done to avoid enzymatic browning. After freezing the structure of the fruit tissue is destroyed due to the presence of the ice crystals. And I insist, especially if one of the main parameters measured was the color.

Authors: Since the sample storage was carried out in order to facilitate the different experiments and prepare the samples for the thin-layer drying in RW™ dryer, these samples were vacuum heat-sealed packaged and frozen storage.

Reviewer: Line 111: cm2 must be in superscript.

Authors: We have corrected the unity superscript.

Reviewer: The following part of the question is not answered: “The weight changes were measured only at before and after drying process? Did you measure during the drying process?.

Authors: We have corrected the sentence.

Reviewer: Lines 108-109: You mentioned that drying times are shorter than air-convective drying, however drying times are not described in the manuscript.

Authors: We have included some references to compare the process times between air-convective drying and both RW™ drying process (non-assisted and IR-assisted). Where we can see that our study gets lower drying times than them.

Reviewer: Equation 2. Why did you work with ratios of moisture??

Authors: When the different drying experiments are performed, and as a measure to standardize the drying experiments (independent of initial moisture content of each sample), the moisture ratios are used. Also, all of the drying data starting from the value of 1.0, this also is important because we can use diverse mathematical models.

Submission Date                    12 February 2020

Date of this review                06 Mar 2020 19:35:56
